

# On the resolutions of ocean altimetry maps

Maxime Ballarotta[1], Clément Ubelmann[1], Marie-Isabelle Pujol[1], Guillaume Taburet[1], Florent Fournier[1], Jean-François Legeais[1], Yannice Faugere[1], Antoine Delepoulle[1], Dudley Chelton[2], Gérald Dibarboure[3], Nicolas Picot[3]

[1]Collecte Localisation Satellite, Ramonville-Saint-Agne, 31520, France
[2]College of Earth, Ocean and Atmospheric Sciences, Oregon State University, Corvallis, OR
[3]Centre National d'Etudes Spatiales, Toulouse, 31400, France

*Correspondence to*: M. Ballarotta (mballarotta@groupcls.com)

**Abstract.** The DUACS system produces sea level global and regional maps that serve oceanographic applications, climate forecasting centers, geophysics and biology communities. These maps are constructed from optimal interpolation of altimeter observations and are provided on a global ¼°x ¼° (longitude x latitude) and daily grid resolution framework (1/8°x1/8° longitude x latitude grid for the regional products) through the Copernicus Marine Environment Monitoring Service (CMEMS). Yet, the dynamical content of these maps is not ensured to have a full ¼° spatial and 1-day resolution, due to the filtering properties of the optimal interpolation. In the present study, we estimate the "effective" spatial and temporal resolutions of the newly reprocessed delayed-time DUACS maps (aka, DUACS-DT2018). Our approach is based on the spectral coherence between maps and independent datasets (along-track and tide gauge observations), which represents the correlation between two sea level signals as a function of wavelength. We found that the spatial resolution of the DUACS-DT2018 global maps based on sampling by three altimeters simultaneously ranges from ~100km-wavelength at high latitude to ~800km-wavelength in the Equatorial band and the mean temporal resolution is ~28 days period. The mean effective spatial resolution at mid-latitude is estimated to ~200km. The mean effective spatial resolution is ~120 km for the regional Mediterranean Sea product and ~140 km for the regional Black Sea product. An inter-comparison with former DUACS reprocessing systems (aka, DUACS-DT2010 and DUACS-DT2014) highlights the progress of the system over the past 8 years, in particular a gain of resolution in highly turbulent regions. The same diagnostic applied to maps constructed with two altimeters and maps with three altimeters confirms a modest increase of resolving capabilities and accuracies in the DUACS maps with the number of missions.

## 1 Introduction

The Data Unification and Altimeter Combination System (DUACS) generates, as part of the CNES/SALP project and the Copernicus Marine Environment and Monitoring Service (CMEMS), delayed-time (DT) multi-mission altimeter Sea Level Anomaly (SLA) Level 3 (along-track cross-calibrated) and Level 4 (multiple sensors merged as maps or time series) products. A full reprocessing of these products is carried out approximately every 3 years and covers the period 1993 – now. The





reprocessing benefits from improvements associated with optimized mapping parameters and new altimeter corrections which are based on standards recommended for altimeter products by the different agencies and expert groups (Ocean Surface Topography Science Team (OSTST), the ESA Quality Working groups and the ESA Sea Level Climate Change Initiative project members). The former reprocessing was released in 2014 (DUACS-DT2014, see Pujol et al., 2016) and the new release,

namely DUACS-DT2018, is available since April 2018 (Taburet et al., submitted).

The Level 4 DUACS-DT global maps are constructed from optimal interpolation (Bretherton et al., 1976, Le Traon, 1998, Ducet et al., 2000) of Level 3 altimeter observations and are provided on a regular ¼°x ¼° longitude x latitude and daily grid resolution framework (1/8°x1/8° horizontal sampling for the regional Mediterranean and Black Sea products). However, the optimal interpolation used in DUACS does not allow the restitution of the full dynamical spectrum of the ocean, limiting the

capability of retrieving small mesoscale in Level 4 products (Chelton et al, 2011 and 2014).

The "effective" resolution corresponds to the spatio-temporal scales of the features that can be properly resolved in the maps. The spatio-temporal resolution of the former Level 4 global SLA products was estimated by Chelton et al. (2003, 2011, 2014) based on estimates of the mapping errors in Sea Surface Height (SSH) fields constructed from altimeter data, or spectral ratio analysis between maps and along-track. Their analysis suggested mid-latitude spatial resolution capability of the observations

ranging from ~2° to 6°, depending on the number of altimeters used in the merging and the sampling pattern of the ground track (~2° for tandem mission T/P-Jason 5 days offset between parallel tracks, 6° for T/P mono-mission merging). The temporal resolution capability of the observations was estimated ~20 days.

In the present study, we further investigate the effective resolution of the DUACS-DT gridded products using a cross-spectral approach. The objective of the paper is threefold: 1) to deliver the spatial distribution of the effective resolutions as key

information to the users about the quality and the limitations (in term of resolution) of the newly produced DUACS-DT2018 gridded products, 2) to access and compare the spatial and temporal resolution capabilities of the DUACS-DT2018, DUACS-DT2014 and DUACS-DT2010 maps (i.e., to identify the impact of system upgrades), and 3) to verify the impact of the varying satellite constellation on the effective resolutions of the maps. The paper is organized as follow: the data and method are introduced in section 2. In section 3, we present our results. Finally, a discussion and a conclusion are provided in section 4.

A detailed description of the choices and method is given in Appendix A.

## 2 Data and method

### 2.1 Input data

In the present study, we consider two kinds of data:

- Independent dataset: we used two independent (i.e., not used in the mapping) datasets to evaluate the effective
resolutions of the maps:  1) Level 3 SLA from independent 1Hz along-track and 2) the SLA estimated from tide gauges. The along-track SLA are constructed using a procedure similar to Level 3 CMEMS products and are used to estimate the effective spatial resolution. The SLA at tide gauge locations originate from the Global Sea Level



Observing System and Climate and Ocean Variability, Predictability and Change (GLOSS-CLIVAR) network and is used to estimate the effective temporal resolution. GLOSS-CLIVAR data are worldwide and available with daily sampling.

• The maps of SLA are constructed using optimal interpolation, based on the a priori statistical knowledge of the field (e.g., variance, correlation scales, noise). The mapping procedure is based on merging of calibrated multi-satellite altimeter (Level 3) data and follows the same protocol as described by Pujol et al. (2016) for the DUACS DT2014. Taburet et al. (submitted) give the full description and validation of the DUACS-DT2018 global and regional products. The main differences between the DUACS-DT2014 and the DUACS-DT2018 processing consist of an
improved along-track processing (e.g., improved orbit correction, wet troposphere correction, ocean tide correction and a new mean sea surface) and updated a priori knowledge of the SLA variance and optimized selection of the data in the optimal interpolation. The maps tested here are computed specifically for this study in several constellation scenario, keeping at least 1 mission out to allow an independent assessment of the resolution. The DUACS-DT products, formerly known as AVISO products, are referenced in the CMEMS catalogue as "OCEAN GRIDDED L3/4
SEA SURFACE HEIGHTS AND DERIVED VARIABLES REPROCESSED" products.

## 2.2 Method

Our method to estimate the effective resolution is based on the "Magnitude Square Coherence" (MSC) between mapped SLA signal and the signal originating from the independent datasets previously mentioned. The MSC represents the squared correlation between two signals in the spectral domain. We somewhat subjectively define the effective resolution to be the
wavelength above which the MSC exceeds 0.5. The principle of the method follows three main steps as illustrated in Figure 1: 1) data selection over common period and area, 2) colocation of SLA signal between independent dataset and maps of SLA and 3) cross-spectral analysis between the two SLA signals. A full description of the method is given in Appendix A.

To justify the choice of the 0.5 criterion for the estimation of the resolution, we follow an explanation given by Bendat and Piersol (1986), and specifically, their case study #2 (No output noise; uncorrelated input noise) in their section 6.1.3. Like their
case study #2, we suppose, in our analysis, that the gridded SLA signal (MSLA = y(t)) contains no noise (Eq. 2) and that the along-track SLA (SLA=x(t)) is the sum of a signal $u(t)$ and a noise $m(t)$ that is assumed to be uncorrelated with $u(t)$ (Eq. 1). Auto-spectral $G_{xx}$ (Eq. 3), $G_{yy}$ (Eq. 4) and cross-spectral $G_{xy}$ (Eq. 5) densities are then expressed as:

$$SLA(t) = x(t) = u(t) + m(t) \quad (1)$$
$$MSLA(t) = y(t) = v(t) \quad (2)$$
$$G_{xx}(f) = G_{uu}(f) + G_{mm}(f) \quad (3)$$
$$G_{yy}(f) = G_{vv}(f) \quad (4)$$
$$G_{xy}(f) = G_{uv}(f) \quad (5)$$



Based on these assumptions, they demonstrate that the MSC (Eq. 6) between $x(t)$ and $y(t)$ is:

$$\gamma_{xy}^2(f) = \frac{\left|G_{xy}(f)\right|^2}{G_{xx}(f)G_{yy}(f)} = \frac{1}{1 + \left|\frac{G_{mm}(f)}{G_{uu}(f)}\right|} \quad (6)$$

Finally, it follows that when the along-track noise-to-signal spectral ratio $\left|\frac{G_{mm}(f)}{G_{uu}(f)}\right|$ is equal to 1, the coherence function $\gamma_{xy}^2$ is equal to 0.5. In other words, the 0.5 criterion corresponds to the level where there is an equal amount of signal and noise in the

reference along-track data. It is the same definition that is used in Dufau et al., (2016) to estimate what they call the ''1-D mesoscale resolution capability, i.e., the minimum size of dynamical structures that altimetry would statistically be able to observe". It is worth mentioning that, although our definition is based on the traditional threshold to estimate along-track resolution in the spectral domain (e.g., as in Yale et al., (1995); Smith (2015); Dufau et al., (2016)), it differs from criterion chosen in the spatial domain analysis by Chelton et al. (2018). To illustrate and discuss the impact of the choice of the Signal-

to-Noise Ratio (SNR) criterion on the resolution, a sensitivity study is provided in the Appendix C. We demonstrate that the resolution can be ~20% coarser with SNR = 4 and > 20 % coarser with a more conservative SNR criterion (e.g., SNR=10, as recommended by Chelton et al, 2018).

## 3 Results

### 3.1 Effective resolutions of the DUACS-DT2018 maps

The effective spatial resolution of the DUACS-DT2018 global maps is shown in Figure 2a. Resolution was computed for maps constructed with three altimeters (Cryosat-2, HY-2, Jason-2) over the period 20140412-20151231 and Saral/Altika data were used as an independent dataset. We believe that this assessment of the spatial resolution based on maps constructed with three altimeter missions may be considered as a reasonable averaged estimation since ~3 altimeter missions are used in the merging for the CMEMS products 70% of the time over the period 19930101-20170515. The resolution ranges from ~100 km

wavelength at high latitudes to ~800km wavelength near the Equator, with a mean resolution at mid-latitude near 200km. Considering that eddy radius characteristic can be estimated as 20-25% of the wavelength (Chelton et al., 2011; 2018), this means that ~25km radius structures are properly resolved in the maps at high latitudes, ~200km radius structures are resolved in the Equatorial band and ~50km radius structures are resolved at midlatitudes. The effective spatial resolution of the DUACS-DT2018 Mediterranean Sea maps ranges from 90 to 200 km wavelength (Figure 2b). The averaged resolution is ~118 km

wavelength over the basin. The effective spatial resolution of the DUACS-DT2018 Black Sea maps ranges from 100 to 200 km wavelength and the averaged resolution is ~136km wavelength over the basin (Figure 2b).

The effective temporal resolution of the DUACS-DT2018 maps ranges from 13 to 49 days period (Figure 3). The temporal resolution is heterogeneously distributed over the global ocean, particularly in the inter-tropical band where a wide range of scales are found, linked to the mixture of continental tide gauges and island tide gauges, with the latter being more



representative of open-ocean conditions. At mid-to-high latitudes the temporal scales are between 14- and 28-day periods, coherent with the temporal correlation scales applied in the mapping process. The globally averaged effective temporal resolution is estimated ~28 days period.

The globally averaged resolutions of about 200 km by 28 days period are consistent with the resolutions reported by Chelton
et al. (2011; 2014) and Pujol et al. (2016). Using the spectral ratio method, they found spatial resolution slightly less than 200 km at mid-latitude in Pacific Ocean.

## 3.2 Evolution of the DUACS system

We here investigate the impact of the DUACS system upgrade from 2010 to 2018 to highlight the progress of the DUACS processing. Resolutions were computed for maps constructed with two altimeters (Topex-Poseidon and Jason1) over the period
20030101-20041231 and Geosat Follow On data were used as independent dataset. To identify the impact of the DUACS system upgrade, we computed the relative improvement/deterioration of the effective resolutions (expressed in percentage) for the upgrade DT2010 to DT2014, and DT2014 to DT2018 (Figure 4). Negative (positive) value means finer (coarser) resolution with the upgrade. The comparison of the DT2010 and DT2014 processing shows finer resolution (improvement > 2%) in the DT2014 than in DT2010 in the high variability region, e.g. the Gulfstream system, the Kuroshio system and the Antarctic
Circumpolar Current (ACC) (Figure 4a). These improvements are associated with updated processing such as updated instrumental and atmospheric correction, tide correction, inter-calibration method and smaller correlation scale in the mapping process. Coarser resolutions in DT2014 than in DT2010 are found in the Equatorial band and are potentially linked to larger correlation scales applied in this region in the DT2014. Although the DT2018 and DT2014 global maps have similar mean effective spatial resolution, regional investigation highlights ~2 to 10% improved resolution in DT2018 in highly turbulent
region (Figure 4b), such as the Equatorial region, the Gulfstream system, the Kuroshio system as well as some regions in the ACC. These improvements are linked to the new mapping standard (optimized selection of the observations in the turbulent region and a priori knowledge of the SLA variance based on a longer period in DT2018). The loss of resolution in the South Equatorial Atlantic is not understood yet.

Similar comparison is performed for the Mediterranean and Black Sea regional products focusing on the upgrade DT2014 to
DT2018. Resolutions were computed for regional DUACS maps constructed with three altimeters (Jason-2, Cryosat-2, HY-2) over the period 20140412-20151231 and Saral/Altika was used as an independent dataset. The resolution capability of the Mediterranean Sea maps is slightly finer (~4%) in DT2018 than in DT2014 (Figure 4c). The largest improvements (>6%) are found the western Mediterranean basin. The resolution in DT2018 is slightly coarser in the Aegean Sea and along the Costa del Azahar.  In these regions, the limited number of along-track data restricts a reliable interpretation of the spectral signal (see
Figure A1). The resolution capability of the Black Sea maps is on average slightly finer (~3%) in DT2018 than in DT2014, although a deterioration is found in the central part of the basin, which is also linked to a reduced number of spectral computations (Figure 4c).



The DUACS-DT2018 and DUACS-DT2014 maps have similar mean effective temporal resolution of ~ 28 days period (globally averaged difference < 4%). The differences can be locally larger than 15%, near 30° along the Japanese coast (~10 days gain), as shown in Figure 5. In these regions, the temporal resolution in the DUACS-DT2018 is finer than in DUACS-DT2014. These regions also coincide with the largest increased correlation score in the DUACS-DT2018 between SLA time series from maps and from independent tide gauge sensors (Taburet et al., submitted). These coastal improvements are linked to the new altimeter standards in coastal regions in DT2018 (Taburet et al., submitted).

### 3.3 Impact of altimeter constellation on the effective spatial resolution

Since the number of altimeter data processed by the DUACS system varies with time (according to the availability of satellites and the data quality), we investigated the impact of the constellation on the effective spatial resolution. Figure 6 illustrates the impact of the number of altimeters (2 or 3 missions) used in the mapping on the effective spatial resolution. We verify, with our diagnostic, modest increases of resolving capabilities in the DUACS maps with increasing number of altimeters and found a globally averaged gain of resolution of ~5% from maps constructed with two altimeters to three altimeters. Regional gains of resolution can be larger than 10%. Additionally, it is possible to identify the improved resolving capability when a mission is introduced in the DUACS system: for example, Figure 6a illustrates the improved resolving capability when mission HY-2 is introduced in the mapping, Figure 6b illustrates the improved resolving capability when mission Cryosat-2 is introduced in the mapping. It is shown that the major contribution of the HY-2 mission in the mapping is in the high variability regions (Gulfstream, Kuroshio, Agulhas systems) while Cryosat-2 contributes in the mid-to-high latitude regions. On the global scale, the distribution of the effective spatial resolution is shifted toward shorter scales when the number of missions used in the merging increases (Figure 7) or when recent altimeters are used in the interpolation (e.g., compare the resolution maps from DT2018 constructed with historical Jason-1/Envisat versus the maps from DT2018 constructed with currently operational missions Jason2/HY-2 or Jason-2/Cryosat2).

### 4 Discussion and conclusions

The present study investigates the resolving capability of the DUACS delayed-time gridded products (Global, Mediterranean Sea and Black Sea) delivered through the CMEMS catalogue. The key results are summarized in Table 1. Our method is based on the spectral coherence and is similar to the approach proposed by Yale et al. (1995) or Smith (2015) to estimate along-track resolution. While along-track altimeter data resolve scales in the order of few tens of kilometers (Dussurget et al., 2011, Dufau et al. 2016), we found that the merging of these along-track data into continuous maps in time and space leads to properly resolved structures with a feature radius resolution of 25km (effective resolution of 100km wavelength) at high latitudes to 200km (effective resolution of 800km wavelength) near the Equator in the global gridded product and with a temporal scale of about 28-day period. The same analysis applied to the regional Mediterranean Sea and Black Sea products showed resolving capability of structure with feature radius resolution of ~30 km, which corresponds to ~3 grid spacing.



These results are consistent with previous investigations. Based on a spectral ratio approach (cf. Appendix B), Chelton et al. (2011) estimated a wavelength resolution of ~200 km for DT2010 and Chelton et al. (2014) estimated a wavelength resolution of ~180 km for DT-2014 in the mid-latitude Pacific Ocean. Our analysis based on the DT2018 global maps, suggests a wavelength resolution of ~200 km at mid-latitudes. As illustrated in Figure 8, we verified our estimation of the zonally

averaged feature radius resolution of the mesoscale structures that can be properly mapped is smaller the eddy scales computed by Chelton et al. (2011). The eddy length scales range from ~70 km at high latitudes to ~180km near the Equator. The effective resolution is ~1.6 smaller than the eddy length scale. Additionally, we confirm that the minimum 4 weeks lifetime criteria used by Chelton et al. (2011) to identify and follow eddies seems to be reliable. Note that our time scale estimation is based mainly on coastal locations and might not be representative of all oceanic regime.

The comparison of the DUACS-DT2018 reprocessing with former DUACS reprocessing (DT2010 and DT2014) reveals that finer structures are globally mapped in the global and regional Mediterranean Sea DT2018 products. For the Black Sea product, the interpretation is more complex due to the small dimension of the basin and the limited amount of spectral computation. Globally, we found that the largest improvements reach 20% and are mainly in high variability regions, associated with the new mapping standard (e.g., optimized selection of the along-track data, new a priori knowledge of the signal variance based

on 25 years of altimetry data, updated correlation scales for the regional Mediterranean Sea product) and new altimeter standards (e.g., instrumental and atmospheric correction, tide correction, inter-calibration method). The improvement patterns between DT2014 and DT2010 global maps is similar to those found by Pujol et al. (2016) using statistical comparison between maps and independent along-track, and drifters' datasets. The improvement patterns between DT2018 and DT2014 global maps coincide with those found by Taburet el al. (submitted) for the validation of the DT2018 products. Using statistical

comparison between maps and independent along-track, Taburet et al. (submitted) also found improvement (~3-4%) of the mapped mesoscale structures, in the high variability region and in the western Mediterranean Sea basin. Note that, at global scale, Taburet et al. (submitted) diagnosed the largest improvements between DT2018 and DT2014 in the coastal regions which are partially edited (along 100km coastal band) in the processing for estimating the spatial scale in this study, but they are detected with the temporal scale analysis, showing shorter timescale in the DT2018 compared with DT2014.

Several studies showed that at least two altimeters are required to accurately map the SSH mesoscale structures (Le Traon and Dibarboure 1999; Ducet et al. 2000; Pujol and Larnicol 2005; Dibarboure et al. 2011; Chelton et al. 2007; 2011) and up to four altimeters are required for Near-Real-Time products (Pascual et al. 2006). The present study reinforces these findings, showing that the resolution capability increased ~10-20% at regional scale from two to three altimeters merging.

It is worth noting, that we probably underestimate the resolution capability of the maps since we are estimating the spatial

effective resolution of degraded maps to keep an independent dataset aside. The resolution might hence be somewhat finer in the distributed CMEMS products. Although the satellite constellation ranges from 1 to 5 altimeter(s) between 19930101 and 20170715, we believe that our estimation of the spatial resolution based on maps constructed with three altimeter missions may be considered as a reasonable averaged estimate since ~3 altimeter missions are used in the merging for the CMEMS products 70% of the time over the period 19930101-20170515. We can expect > 5% finer resolution over period where more





than 4 altimeters are available (i.e., recent period). Likewise, we can expect on average 5% coarser resolution when only 2 altimeters are available.

To conclude, the number and the quality of altimeter simultaneously operational, the along-track configuration and sampling pattern, the weight given to the altimeter data in the mapping procedure and the choice of threshold MSC or SNR (see Appendix
C) are key factors controlling the resolution capability of the DUACS gridded products. One may expect that, in permitting to observe finer mesoscale/sub-mesoscale structures (Dufau et al., 2016, Pujol et al, 2012), future instrumental systems based on large-swath altimeter (such as Surface Ocean and Water Topography (SWOT)) combined with new mapping technique based on dynamic interpolation (Ubelmann et al., 2016), will push the maps resolution toward new limit.

*Code and data availability.* The DUACS system source code is not publicly available. The code for the spectral analysis is released under GNU General Public License v3.0 and is available at https://github.com/mballaro/scuba. DUACS all satellites gridded data and along-track data are available through the CMEMS website: http://marine.copernicus.eu/. Specific maps used in our study are based on merging of three or two satellites and are available on request by contacting M. Ballarotta (mballarotta@groupcls.com).

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

**Appendix A: Full description of the method**

Definition of the effective spatial resolution

The diagnostic to estimate the effective spatial resolution is based on the calculation of the magnitude square coherence (MSC), which measures the phase and amplitude consistency (i.e., the correlation) of two SLA signals as a function of the wavelength (Eq. 7). It is defined as follows:

$$MSC(\lambda) = \frac{|CS_{xy}(\lambda)|^2}{[S_x(\lambda)S_y(\lambda)]} \quad (7)$$

, where $x$ denotes the SLA of the independent along-track data, $y$ denotes the SLA of the map interpolated onto the independent along-track segment, $CS$ is the cross-spectral density, $S$ is the power spectral density, $MSC$ is the magnitude square coherence, and $\lambda$ the spatial wavelength.

Methodology: The algorithm to compute the spatial effective resolution follows 4 main steps:

- A coastal editing is applied in a 100 km coastal band (only for the global products) to remove the increased errors in
the coastal area.

- Gridded data are interpolated to the locations of the independent along-track data.

- Along-track and interpolated data are divided into overlapping 1500km long segments every 300 km for the global products (500km long segments for the Mediterranean Sea products and 300km long segments for the Black sea products). Each segment is saved in a database and referenced by it median (longitude, latitude) coordinates.

- Finally, between latitudes 90°S-90°S and longitudes 0-360°E, we consider 10°x10° longitude x latitude boxes for the global products (5°x5° longitude x latitude boxes for the Mediterranean Sea product, and 3°x3° longitude x latitude boxes for the Black Sea product) every 1° incremental step. All available segments referenced within the 10°x10° box are selected to compute the power spectral densities and cross spectral density based on the Welch method (1967). Prior to spectral computation, signals are detrended and we applied a Hanning window. The effective resolution is
then given by the wavelength where the MSC is 0.5.

The spectral coherence approach (applied to the altimetry product) is illustrated in Figure 1 with the data selection and interpolation step to the spectral analysis. The total number of averaged segments in each 1°x1° longitude x latitude box is shown in Figure A1a) for the global product, Figure A1b) for the Mediterranean Sea product and the Black Sea product. Due to the coastal editing, the number of computed segments in the global product analysis is less than 1000 near the coast and
~1500 in the open ocean. In the Mediterranean Sea the number of segments is ~400 and ~250 for the Black Sea. A limitation of the present spectral approach is the need to rely on coastal for estimation of the resolution in the two regional products.



Definition of the effective temporal resolution

A comparison of SLA maps with independent tide gauges dataset is carried out to estimate the effective temporal resolution. The approach is like the estimate of the effective spatial resolution and based on the computation of the magnitude square coherence (Eq. 8):

$$MSC(T) = \frac{|CS_{xy}(T)|^2}{[S_x(T)S_y(T)]} \quad (8)$$

, where $x$ denotes the SLA of the independent tide gauge sensor, $y$ denotes the SLA of the map interpolated onto the independent along-track segment, $CS$ is the cross-spectral density, $S$ is the spectral density, $MSC$ is the magnitude square coherence, and $T$ the temporal period.

We computed the effective temporal resolution from each tide gauge time series of the GLOSS-CLIVAR network. The
temporal domain covers the period 19930101 - 20151231. The computation for each time series follows 3 main steps:

At each tide gauges location, we extract the gridded SLA time series that is most highly correlated with tide gauges time series (note that the maximum distance separation of the grid point that is most highly correlated with each tide gauge is 100km on average and can be as large as 300km)

1) Each highly correlated time series (based on correlation criterion > 0.8) is subsampled into 100-day segments to compute
the spectral coherence based on the cross-spectral density $CS$ and the spectral densities $S_x$ and $S_y$. The length of each segment must be set when estimating power spectral density using Welch's method. By subsampling into 100-day segments, we limit the frequency range to only periods shorter than 100 days. We performed a sensitivity study on longer segment length (200 days and 300 days periods) and found similar global averaged effective temporal resolution (26 days for 300 days long segment, 27 days for 200 days long segment and 28 days for 100 days long segment). Note that
consideration of time series longer than 100 days reduced the number of realizations because of occasional gaps in some of the data records. This can have an impact on the local estimation of the effective temporal resolution, e.g. from 100-day to 300-day segments we lose some continuous time series that do not contain >=200-day segment. In these cases, the spectral analysis cannot be performed.

2) The effective temporal resolution at each tide gauge location is given by the period where the MSC is equal to 0.5.

3) Note that this estimation of the temporal resolution is subject to an important caveat: the estimation is based mainly on coastal locations which may be polluted by altimetry errors. Additionally, it may be not be fully representative of the temporal resolution of the DUACS maps which combine various oceanic regimes (e.g., coastal, offshore high variability, offshore low variability regimes). Additional investigation using offshore moorings might be valuable to consolidate our findings. Our results may therefore be crude but useful estimates of the temporal resolution.




**Appendix B: other spectral approach**

Effective vs Useful spatial resolution

Chelton et al. (2011, 2014) estimated the resolution of the DUACS DT2014 maps based on the calculation of the ratio between the reference Stammer (1997) along-track spectrum and gridded SLA spectra. Similarly, we here estimate the resolution based
on the spectral ratio between independent along-track and gridded SLA signals (Eq. 9). It is defined as follows:

$$SR(\lambda) = \frac{S_x(\lambda)}{S_y(\lambda)} \quad (9)$$

, where $x$ denote the SLA of the independent along-track, $y$ denotes the SLA of the map interpolated onto the independent along-track segment, $S$ is the spectral density, $SR$ the spectral ratio and $\lambda$ the wavelength. The resolution is given by the wavelength where the spectral ratio is equal to 0.5 and is based on the conventional notion of a filter being characterized by
its half-power filter cutoff wavelength (Chelton et al., 2011). To differentiate it from the effective resolution we named it "useful" resolution: "useful" for verifying the available and realistic amount of energy at a specific wavelength between two signals without considering their phase (e.g., useful for model sensitivity study).

Chelton et al. (2011, 2014) estimated the resolution of the DUACS-DT2010 and DUACS-DT2014 as the wavenumber at which the power is a factor of 2 smaller than the Stammer (1997) spectrum. From their analysis, they estimated spatial resolution of
~2° for the DUACS-DT2010, ~1.7° for the DUACS-2014, and found essentially the same resolution between maps constructed with 2 satellites or 4 satellites.

The MSC and SR methods share the same number of spectrum calculation and number of segments used in the calculation. The SR approach complements the MSC approach since SR focuses on the amplitude whereas the MSC is mainly controlled by the phase between the two signals. The resolution estimated with the SR method is shown in Figure B1a) and the ratio
between the effective and useful resolution is shown in Figure B1b). The useful resolution of the DUACS-DT2018 maps ranges from 100km at high latitude to 500km near the Equator. The ratio effective/useful resolution suggests somewhat finer resolution in the intertropical band using SR approach and somewhat finer resolution at high latitude with the MSC approach. In other words, the amplitude of the mapped SLA spectral content is better in the inter-tropical band than the phase, whereas it is the opposite at high latitude. This feature highlights the difficulty to properly map propagating equatorial waves in the
DUACS system.

The scatter plot of the resolution capabilities estimated with each method is presented in Figure B2 and shows that the two methods are equivalent at mid-latitude, whereas the useful resolution is smaller than effective resolution in the intertropical band.



## Appendix C: Sensitivity to the SNR criterion

We here investigate and discuss the impact of the SNR criterion on the estimation of the effective resolution. SNR criterion is used to define the resolution limit in the along-track dataset. In the present study, we choose the SNR=1 criterion (e.g., equal amount of signal and noise) to define the resolution limit of the along-track data. This value may be considered too generous;

therefore, we present below the effective resolution for three cases, motivated by the analysis performed in the spatial domain by Chelton et al. (2018):

- criterion of SNR=1 corresponds to MSC=0.5
- criterion of SNR=4 corresponds to MSC=0.8
- criterion of SNR=10 corresponds to MSC=0.9

Figure C1a) represents the effective resolution using SNR=1 criterion, Figure C1b) using SNR=4 criterion and Figure C1c) SNR=10. For each panel the resolution becomes finer poleward. The white areas correspond to the regions where the MSC threshold criterion is not achieved. These areas become larger in the inter-tropical region as well as at high latitudes when the

SNR criterion increases. For SNR=10, the resolution in the intertropical band cannot be computed with the MSC method, since the maximum MSC is below 0.9. To further illustrate this, we show an example of spectral analysis at one specific point (lon=259°E, lat=22°S) in Figure C2. The analysis shows that the MSC do not exceed 0.7 in this location. This large-scale low coherency between maps and along-track may be linked to the misrepresentation of the large scale and rapid equatorial waves (e.g., equatorial gravity waves) in the mapping process, which are filtered in the mapping process.

Despite the areas of missing values in Figures C1b) and C1c), we quantify the difference of effective resolution between criterion SNR=1 and SNR=4 (Figure C3) and SNR=1 and SNR=10. The ratio between effective resolution SNR=1 vs SNR=4 shows that the difference is < 20% (~40km) at mid-latitude and <=50% (400km) in the inter-tropical band. The ratio between effective resolution SNR=1 vs SNR=10 shows that the difference is < 40% (~80km) at mid-latitude and >50% in the inter-

tropical band.

In conclusion, we here demonstrate that the choice of the SNR criterion has an impact on the estimation of the resolution. Setting more conservative criterion SNR=4 leads to ~20% coarser effective resolution. The strongly conservative criterion SNR=10 also reveals one of the major caveats in the DUACS maps processing: the poor representation of the large and rapid

scale equatorial circulation. This issue should be addressed in the future version DUACS maps.







**Figure 1: Schematic illustration of the Spectral coherence approach: a) input data selection, b) colocation SLA and gridded SLA and c) cross-spectral analysis showing Magnitude Square Coherence MSC**



**Figure 2: Effective spatial resolution in km of the DUACS-DT2018 maps for a) the Global Ocean product, b) the Mediterranean Sea and the Black Sea products. Unit in km**



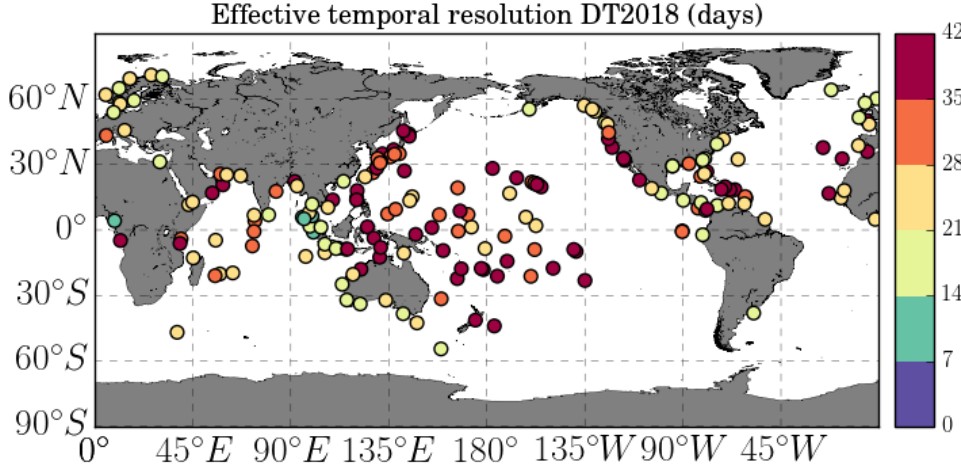

**Figure 3: Effective temporal resolution in days of the DUACS-DT2018 maps. Unit in days**



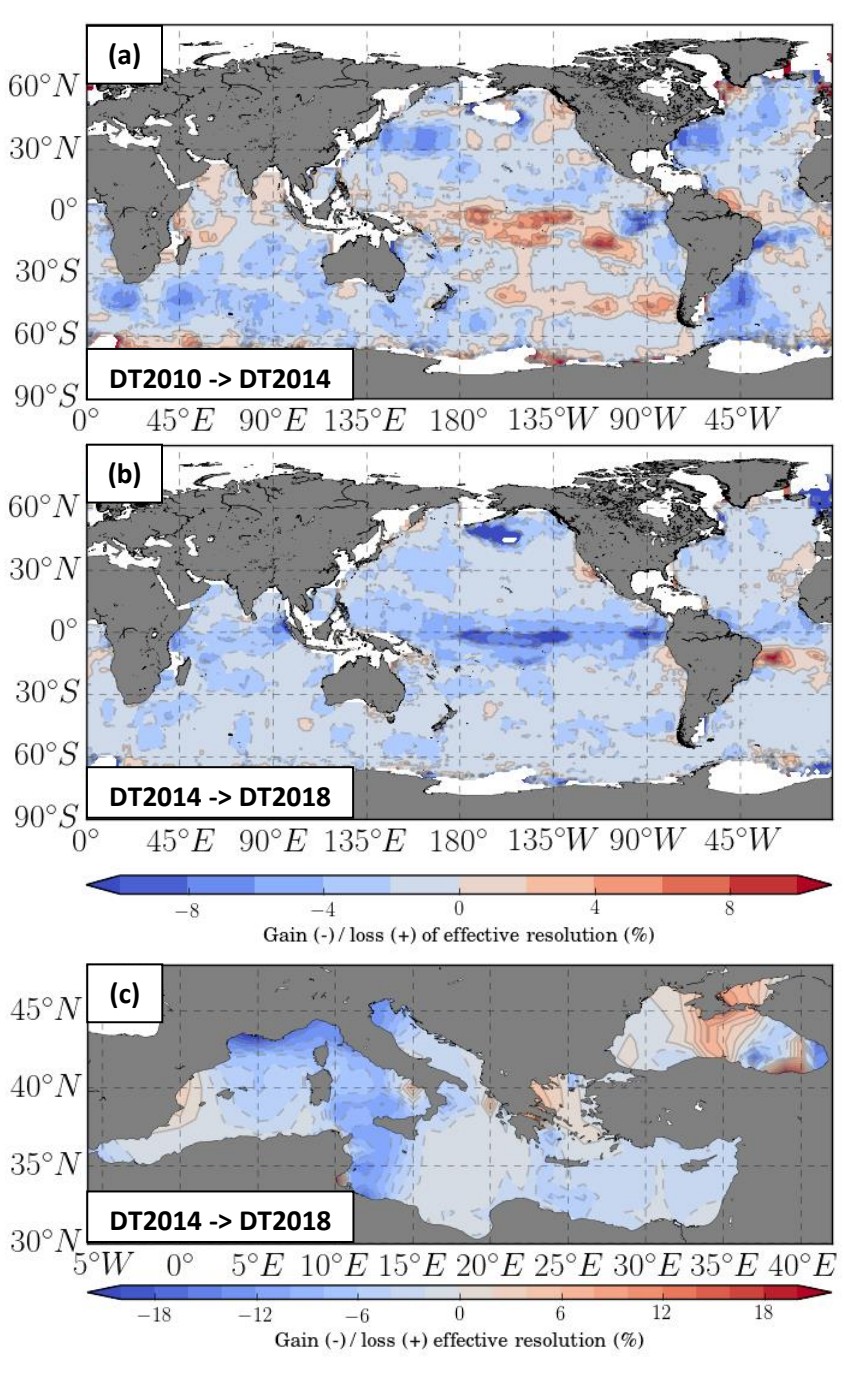

**Figure 4: Gain/loss of effective spatial resolution for a) the Global Ocean product between DT2014 and DT2010, b) the Global Ocean product between DT2018 and DT2014, c) the Mediterranean Sea product and the Black Sea products between DT2018 and DT2014. Negative value means that the resolution capability is finer. Note the different colorbar scale between global and regional products**



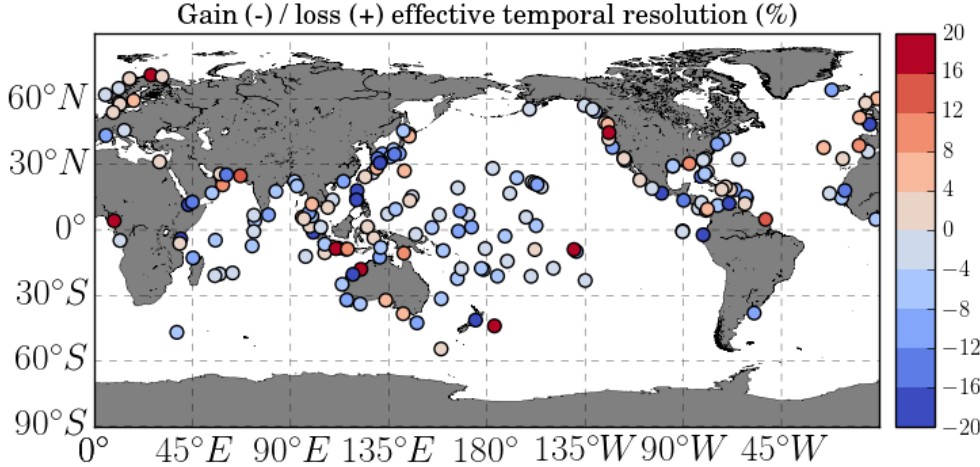

**Figure 5: Gain/loss of effective temporal resolution between DT2018 and DT2014. Negative value means that the resolution capability is better in DT2018 than DT2014**





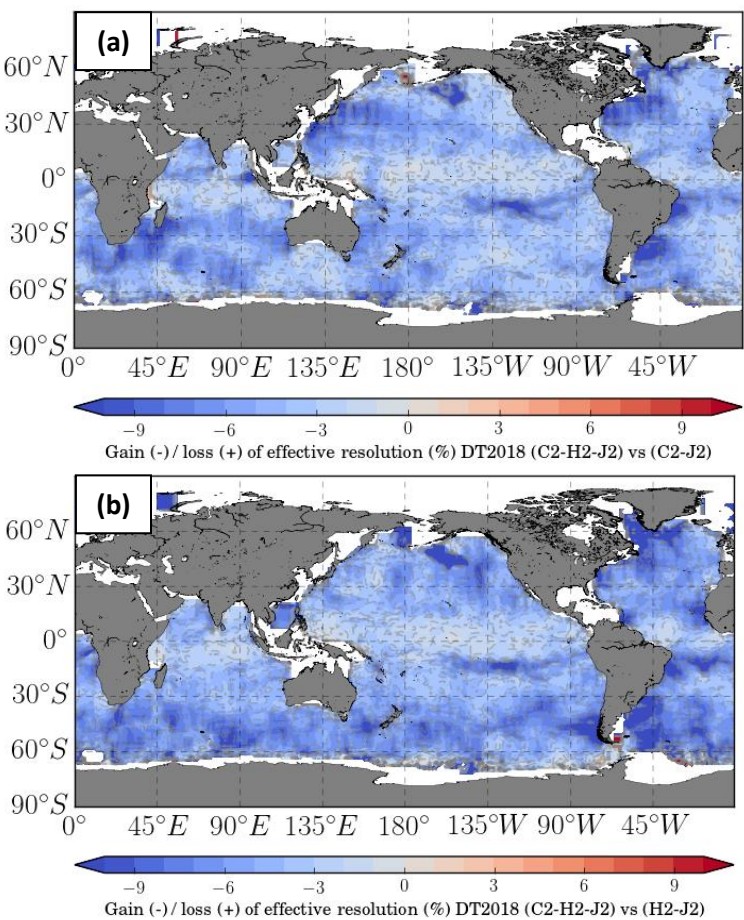

**Figure 6: Impact of the satellite constellation on the effective resolution –
Ratio of effective resolution of a) maps constructed with C2-H2-J2 vs C2-
J2, and b) maps constructed with C2-H2-J2 vs H2-J2**




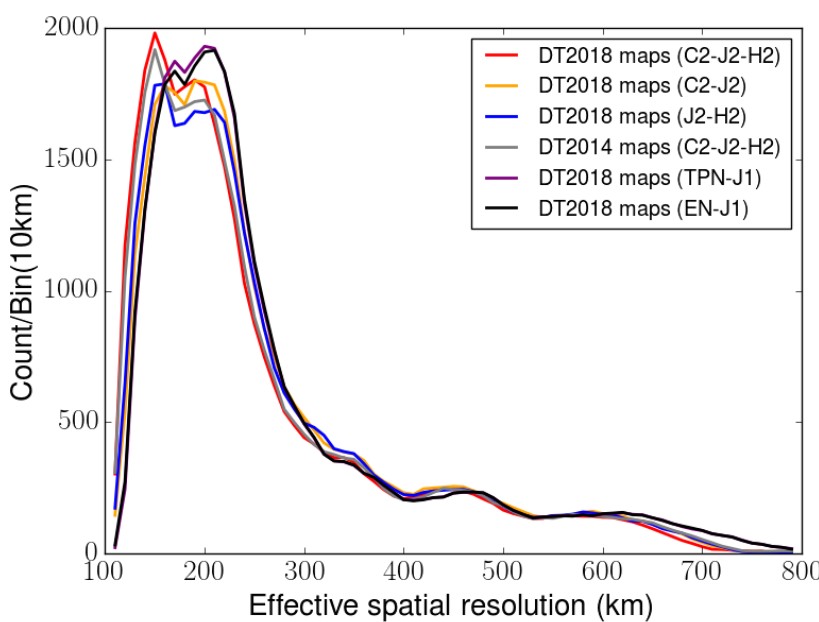

**Figure 7: Distribution of the effective spatial resolution for various altimeter merging configuration**

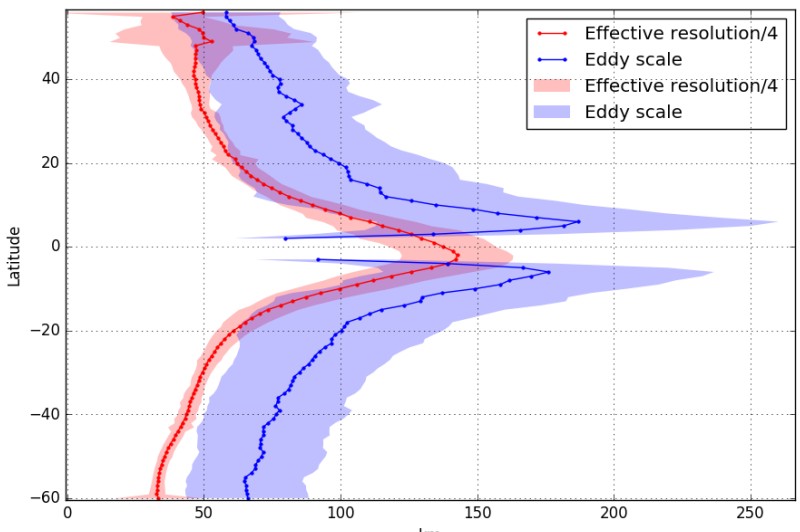

**Figure 8: Zonally averaged eddy scale Ls (as in Chelton et al., 2011; and computed from the DUACS-DT2014 two satellites maps) and feature radius resolution of the mesoscale structures that can be properly mapped in DUACS (i.e., derived as 0.25×effective resolution). Units in km**


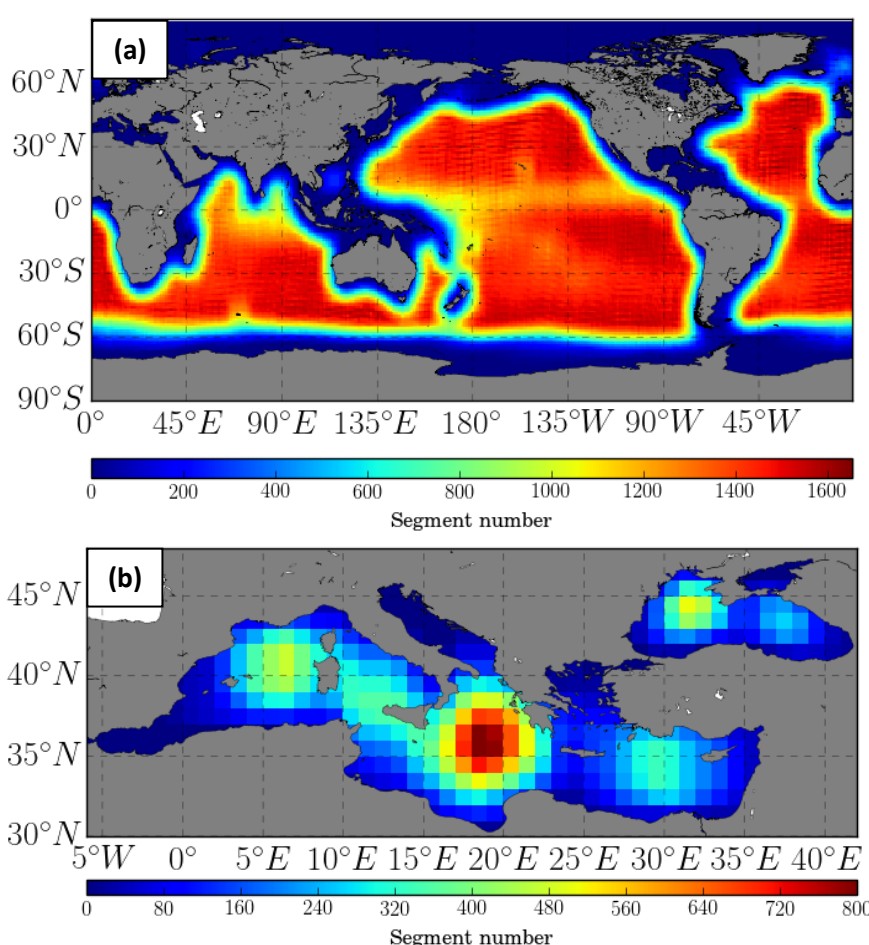

**Figure A1: Number of segments used in the spectral computation for a) the global product and b) the regional Mediterranean and Black Sea products**



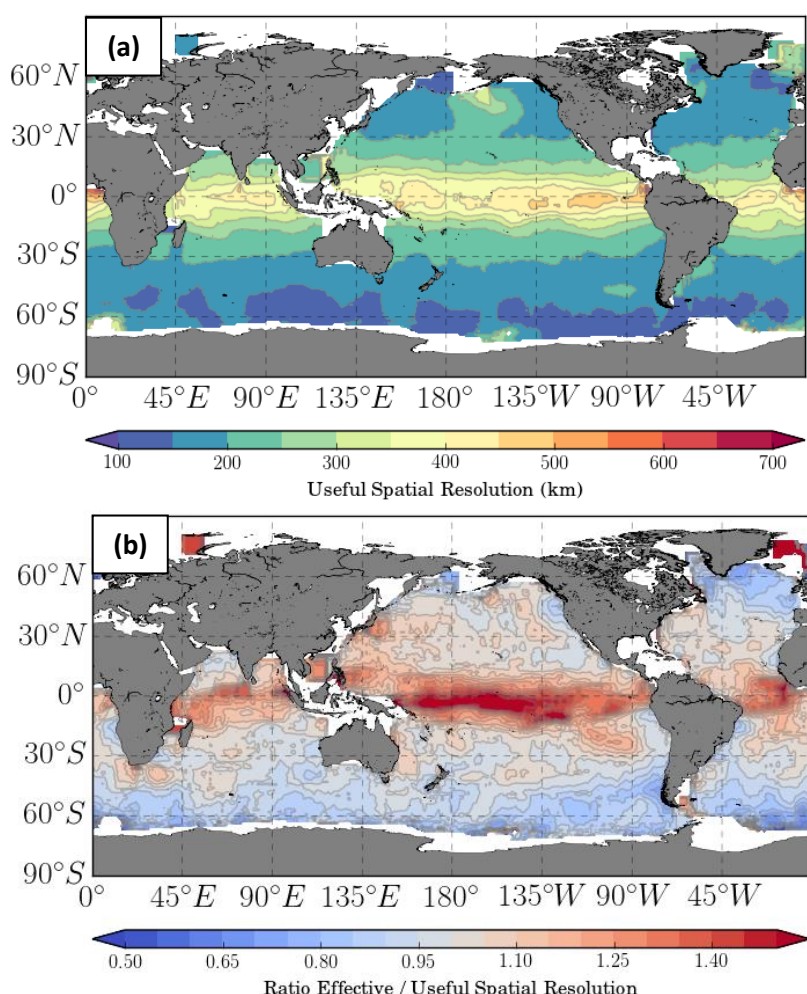

**Figure B1: a) DUACS-DT2018 Useful resolution derived from spectral ratio approach and b) ratio effective / useful resolution for the DT2018 maps**




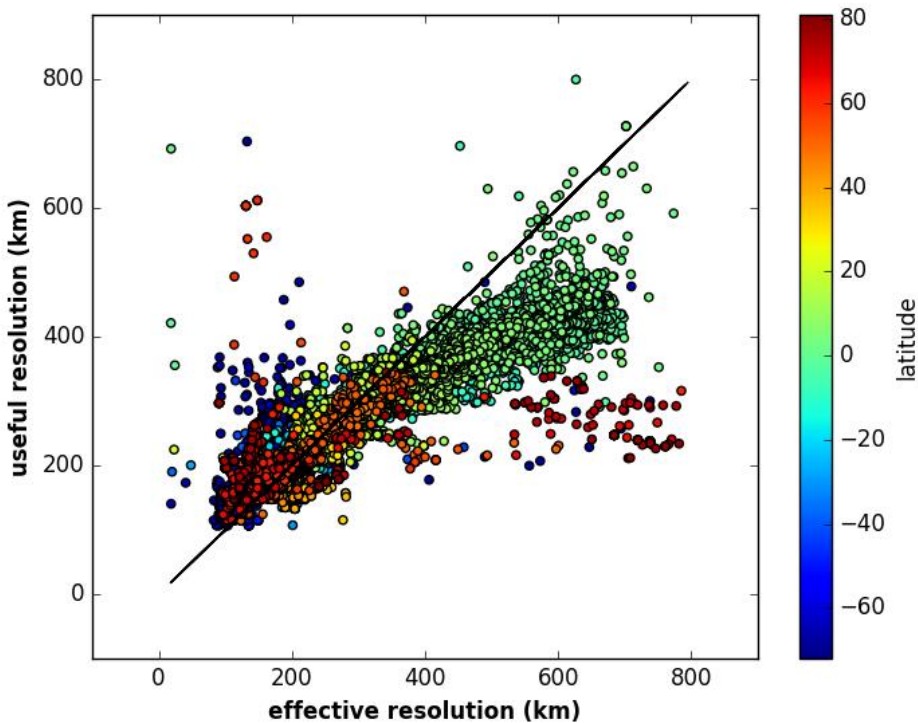

**Figure B2: Scatter plot useful resolution vs effective resolution. Color scale represents the latitude**





30



**Figure C1: Effective resolution computed for three different MSC criteria (0.5, 0.8 and 0.9) corresponding to a) SNR=1, b) SNR=4 and c) SNR=10**



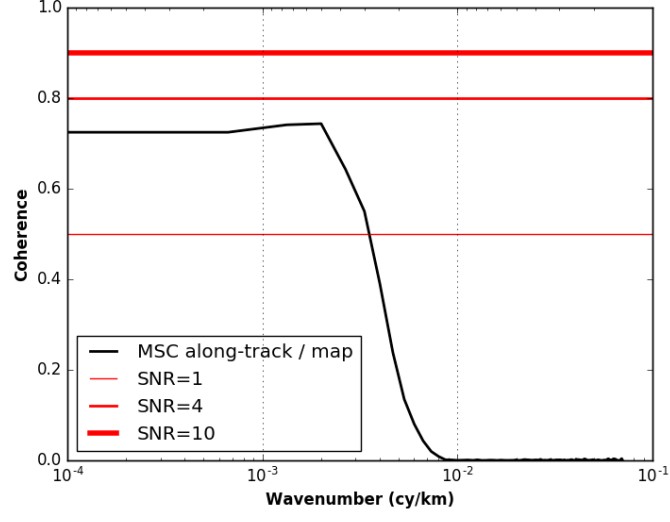

**Figure C2: MSC at longitude 259°E and latitude 22°S. We illustrate that the maximum MSC at this location is always below < 0.8 and so the resolution cannot be computed with MSC criterion > 0.7**

30





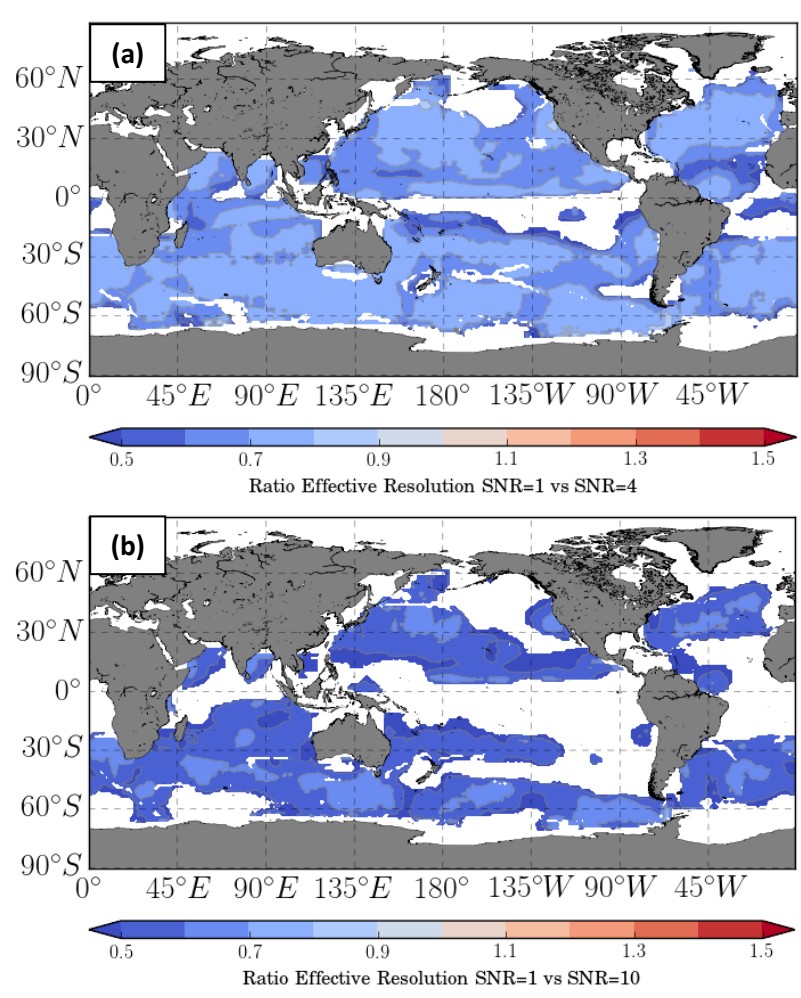

**Figure C3: Ratio effective resolution computed with a) SNR=1 versus SNR=4 criterion and b) SNR=1 versus SNR=10 criterion. Blue means finer resolution with SNR=1**





**Table 1: Summary of the DUACS products spatial and temporal resolutions. (1) Not estimated due to the limited amount of tide gauges in the Mediterranean Sea and Black Sea**

| | SPATIAL FEATURE | | TEMPORAL FEATURE | |
|---|---|---|---|---|
| Product | Effective resolution | Grid spacing | Effective resolution | Grid spacing |
| **GLOBAL** | 100 to 800 km | 4 to 30 km | ~28 days | 1 day |
| **MED-SEA** | ~120 km | ~10km | *(1)* | 1 day |
| **BLACK-SEA** | ~140 km | ~10km | *(1)* | 1 day |

