# Peer review of "On the resolutions of ocean altimetry maps"

_Ocean Science, 2018_

## Referee Comment (RC1) · Lee Lueng (Referee) · 28 Jan 2019

This paper presents a comprehensive study of the resolution of a popular product of ocean altimetry maps issued by the DUACS (AVISO) system. The study used a spectral coherence method to determine the spatial and temporal resolution of the maps, using independent altimetry data (for spatial resolution) and tide gauge data (for temporal resolution). A host of issues have been addressed: the dependence on the number of altimeters, the evolution of the system over the altimetry era, the comparison with previous studies, the sensitivity to the signal/noise ratio. This paper is highly recommended for users of the DUACS maps for any quantitative studies.

Some minor technical comments:

P.1 lines 10-13: The sentence "These maps are ..." is awkward. Needs re-phrasing. P. 2, Line 30: What is the source for the independent 1 Hz along-track data? Although

the answer is provided later, it should be made clear here. P.3, lines 25-30: Why does MSLA contain no noise? Is noise the same as random error? What is t in equations 1-2? what is v(t)? P.4, equation (6): where is Gvv from equation (4)? P.5, lines 17-18: Provide reference on the larger correlation scales. P.7, line 5: add "than" after "smaller". P.7, lines 26-27: Elaborate why up to four altimeters are required for near-real time products. P.7, lines 29-30: This is an important point worth noting when the use of independent track for the study is first mentioned in the paper. P.12, equation (9): either show a figure of SR(lambda) or describe its variation with lambda in words.

---

## Referee Comment (RC2) · Anonymous Referee #2 · 18 Feb 2019

This paper: "On the resolutions of ocean altimetry maps" uses spectral coherence between SSH maps and along-track and tide gauge SSH measurements. The calculations seem correct and the figures are interesting, but to me, this approach is complicated because it combines together the resolution of the maps and the length scales of the processes being imaged. For example, in the equatorial region they estimate spatial resolution at 800km, far bigger than the altimeter track spacing. Chelton, a co-author on this paper, has done a lot of work on evaluating sampling from satellite observations of both SSH and wind (Schlax, et al., 2001), and has examined the transfer function of linear mappings (Schlax and Chelton 1992). The latter paper focuses on the effects of the mapping as a smoother of the original field, examining the resolution of the mapping independently from the length scales of the mapped fields. A similar approach could be used here, taking out the correlation of the underlying fields to focus on the smoothing done by the mapping. This is what I expected from the analysis, and

I feel that the differences should be discussed and the results presented here put in context as a combination of two effects.

I would also like to request that the authors please also address why they do not compute decorrelations in physical space and time instead of coherence. This would allow the preservation of spatial structure that is removed by the stationarity assumption built into the coherence calculation, including the averaging over large regions to get adequate coherence statistics. Since only a decorrelation distance is reported, the sacrifices needed to be able to make the coherence calculation do not seem necessary, and a region-by-region decorrelation scale could have been reported. This would still have mostly represented the scale of the SSH field, not the mapping, but it would be simpler to compute, report, and understand.

references:

Sampling errors in wind fields constructed from single and tandem scatterometer datasets. J. Atmos. Oceanic Tech., 18, 1014-1036, 2001. (Schlax, M. G., D. B. Chelton, and M. H. Freilich.)

Schlax, M. G. and D. B. Chelton, 1992: Frequency domain diagnostics for linear smoothers. J. Amer. Stat. Assoc., 87, 1070{1081.

---

## Referee Comment (RC3) · Tom Farrar (Referee) · 27 Mar 2019

I uploaded my review as a Supplement because I needed to include figures but could not manage to do this well using the LaTeX reviewer interface. Please see the Supplement at the link below.

Please also note the supplement to this comment:
https://www.ocean-sci-discuss.net/os-2018-156/os-2018-156-RC3-supplement.pdf
* * *

---

## Author Comment (AC1) · 23 May 2019

Hi,

We uploaded our comment to the review in the form of a supplement.

Sincerely,

Maxime

Please also note the supplement to this comment:
https://www.ocean-sci-discuss.net/os-2018-156/os-2018-156-AC1-supplement.pdf

---

## Author Comment (AC3) · 23 May 2019

**Author's response to J. Thomas Farrar's Comments on 'On the resolutions of ocean altimetry maps'**

First of all, we would like to thank Tom Farrar for taking time to read our manuscript and for his detailed and inspiring review.

We attach below the entire review made on April 2, 2019. The main concern of the review is the fact the coherence-based measure is not well adapted for estimating the resolution. The review shows several 1D general examples of the behavior of various proposed measures of the resolution (e.g., magnitude squared coherence, filter transfer function), considering low, high noise level and noise free in the input signal. For the 1D example without noise, the squared coherence and filter transfer function highlight different behavior. Similar conclusions are found for the 1D example with low noise level. In a more realistic example with high noise level, the squared coherence gives information about the input SNR but nothing about the filtering.

We agree that the filtering method for similar cutoff wavenumber can impact the coherence amplitude. This impact only happens below the cutoff frequency (Figure 1 in the review) where we should not have any coherence with independent altimetry profiles. If we have some, this means that the DUACS filter has been too aggressive and we may afford shorter correlation scales. In that case, the coherence could be misleading. To avoid that, we can consider evaluating the ratio PSD(SSH_altrack - SSH_map)/PSD(SSH_altrack), i.e., noise-to-signal ratio.

It is important to mention that our computation of the Magnitude Squared Coherence takes only into account the phase consistency between the SSH_altrack signal and the SSH_map signal (not the amplitude). For example, if the correlation scales in DUACS were larger (resulting in smoother SSH, attenuated amplitude), we could have similar phase scores but a poor PSD ratio score. We made the test, artificially smoothing the maps: we obtain the same coherence score but a low PSD ratio score. This convinced us that the coherence is indeed not sufficient as pointed out in the review, but the PSD(SSH_altrack - SSH_map)/PSD(SSH_altrack) score should well characterize the skills, by penalizing both amplitude and phase.

Hence, we propose to switch to the definition of the resolution limit of the maps as **PSD(mapping_error)/PSD(SSH_true) = 0.5** (i.e., the wavelength where the mapping error are two times smaller than the true SSH signal).

In order to illustrate the assessment of the resolution based this new definition, we performed analysis on Observing System Simulation Experiment (OSSE). The details of the simulation and methods are presented in notebooks below; and are freely available / interactively repeatable here: https://mybinder.org/v2/gh/mballaro/notebook.git/master (under the analysis_OSSE_NATL60 folder).

We have performed 3 study cases:

- STUDY CASE #1: Comparison with independent along-track (similar to the approach used in the manuscript)

- STUDY CASE #2: Comparison with NON-independent along-track

- STUDY CASE #3: Comparison with independent along-track with higher instrumental noise that mapping error

The advantage of using OSSE is that we have access to all quantity we want:

- PSD(SSH_map)

- PSD(SSH_true)

- PSD(mapping_error)

- PSD(instrumental_error)

- PSD(SSH_obs) = PSD(SSH_true + instrumental_error)

With real DUACS dataset we can only get:

- PSD(SSH_map)

- PSD(SSH_obs)

- PSD(instrumental_error)

We show in the notebooks that we can approximate the ratio PSD(mapping_error)/PSD(SSH_true) with the three quantities (PSD(SSH_map), PSD(noise_SSH), and PSD(instrumental_error) ).

**PSD(mapping_error)/PSD(SSH_true) = [PSD(SSH_obs - SSH_map) - PSD(instrumental_error)] / [PSD(SSH_obs) - PSD(instrumental_error)]**

The main conclusions from the STUDY CASE #1 notebook are:

- the resolution estimated with the magnitude squared coherence is in good agreement with the PSD(mapping_error)/PSD(SSH_true) = 0.5 approach, linked to the fact that the signal amplitude is globally optimal at the wavelength where the phase becomes incoherent, and thus the major concern for the DUACS system is in the phase consistency between SSH_altrack and SSH_map signal rather than in their amplitude.

- the ratio **PSD(SSH_obs - SSH_map)/PSD(SSH_obs) (blue curve)** and **[PSD(SSH_obs - SSH_map) - PSD(instrumental_error)] / [PSD(SSH_obs) - PSD(instrumental_error)]** (yellow curve) are similar to the **PSD(mapping_error)/PSD(SSH_true) (red curve)** for wavelength > 70km

- the ratio **PSD(SSH_obs - SSH_map)/PSD(SSH_obs) (blue curve) or [PSD(SSH_obs - SSH_map) - PSD(instrumental_error)] / [PSD(SSH_obs) - PSD(instrumental_error)]** (yellow curve) can thus be used to estimate map resolution

- the signals for wavelength < 50 km (grey area) are not meaningful since we are under the grid spacing of the DUACS gridded product as well as in the instrumental noise level

- the sensitivity of the ratio PSD(SSH_obs - SSH_map)/PSD(SSH_obs) to the PSD(instrumental_noise) is weak for wavelength > 70km

Conclusions in STUDY CASE #1 are also valid for STUDY CASE #2.

For STUDY CASE #3 with unrealistic high instrumental noise, the ratio PSD(SSH_obs - SSH_map)/PSD(SSH_obs) becomes extremely sensitive to the instrumental noise level.

Note that this STUDY CASE #3 is not happening in DUACS processing.

In conclusion, we have made the choice to change our definition of the measure of the resolution in the revised version of the manuscript. It was previously based on the Magnitude squared coherence. It is now based on the Noise-to-Signal ratio, which is more robust penalizing both amplitude and phase consistency between the two signals. We illustrate using OSSE that the definition of the resolution based on Noise-to-Signal ratio or the Magnitude squared coherence are equivalent for the DUACS system. The main outcomes of the paper are thus unchanged. We obtain similar spatial and temporal resolutions since the signal amplitude in DUACS is globally optimal at the wavelength where the phase becomes incoherent.

We have thus performed the analysis of the DUACS maps using the NSR ratio measure. The figures have been updated in the revised version of the manuscript. A comparison of various approach to estimate the resolution capability is added and provided in the Appendix A of the paper. It includes spectral magnitude ratio approach and the transfer function approach.

---

## Author Comment (AC4) · 23 May 2019

Hi,

We attach the notebooks in a form of supplement.

Sincerely,

Maxime

Please also note the supplement to this comment:
https://www.ocean-sci-discuss.net/os-2018-156/os-2018-156-AC4-supplement.pdf

---

## Referee Report (RR1)

**Updated review of paper submitted to Ocean Science: "On the resolutions of ocean altimetry maps" by Ballarotta, Ubelmann, Pujol, Taburet, Fournier, Legais, Faugere, Chelton, Dibarboure, and Picot**

J. Thomas Farrar
Department of Physical Oceanography
Woods Hole Oceanographic Institution

April 3, 2019

Note: This was the review of the first version of the manuscript. It was submitted directly to the authors during the revision process and addresses some issues that were not made clear in the first version of the review posted on the OSD website. This is the version the authors responded to in their revision.

This paper presents a new method for assessing the spatial and temporal resolution of a gridded data product and applies the method to the DUACS altimetry product. The paper is well written and the undertaking is worthwhile. The method for assessing resolution is based on estimation of the coherence between independent data and the gridded product. I have a great deal of respect for these authors, and so it is with a lot of discomfort and hesitation that I must say that the method used in the paper is fundamentally flawed, and I strongly recommend against publication. In what follows, I will focus on what is wrong with the proposed method of assessing resolution, because that is essentially the only problem I have with the paper. I would like to express my sincere regret to the authors that I must argue so strongly against publication of their paper– I hope they find this review helpful rather than offensive. I also apologize to all involved for the lengthy review! I felt it was necessary to include some examples to illustrate what is wrong with the methodology. If my assessment is incorrect, I would be happy to be corrected. (In order to begin to convince me, the authors would need to first produce an example in 1-D of how the coherence-based measure of resolution can quantify the filtering properties of a filtering operation performed on the data– as shown in Equation 6 of the paper and below in Equation 14, this looks like an impossible task.)

The method is based on a conceptual model of a linear single-input/single-output system (in the terminology of the book by Bendat and Piersol that the authors cite) that is used to interpret the coherence of the data product with independent data. The conceptual linear system has no noise on the measurement of the system output but does have noise on the measurement of the system input. As described by the authors and by Bendat and Piersol (2010, p. 185), there is an input signal u(t) that goes through a linear system (the mapping) to produce an output signal y(t). The output y(t) is known perfectly (without measurement noise), but the input u(t) is not known perfectly– only a noisy measurement of it is available, x(t)=u(t)+m(t). For brevity, I will refer to this as a "Case 2" model for interpretation of the coherence. The proposed measure of resolution is the wavenumber (or frequency) at which the coherence of the mapped field with an independent 1-D data record is equal to 0.5; in the Case 2 model, this is the wavenumber at which the signal-to-noise ratio is equal to 1. In my earlier version of this review, I may have been confused about which signal the authors consider as the 'input' and which they consider as the

'output'– this distinction does not really matter for the arguments I present below.

The resolution capability of a mapped field should depend on (i) the noise and signal levels in the input data, (ii) the sampling of the data, and (iii) the manipulations performed on the data during the mapping (e.g., filtering). The coherence method addresses (i) but not (iii). It isn't clear whether the method addresses (ii), but the fact that the method can tell us nothing about (iii) means it cannot be a useful measure of resolution by itself.

There are two problems:

1. The most basic problem is that, in 1-D, the coherence should be completely unaffected by the mapping and its filtering and instead depends only on the signal-to-noise ratio of the raw input data. This can be clearly seen from Eqn 6 of the paper, but I have included a derivation below to make this more clear. The derivation shows that the method cannot yield any useful information about the filtering in a 1-D example. The method can give us information about the SNR in the input data, but not about the filtering.

2. A secondary problem arises when the method is applied by computing the coherence between a 1-D sample of SSH against the mapped field to estimate the resolution in that direction. Take the example of computing coherence between independent along-track data and the mapped SSH. The along-track coherence is affected by across-track filtering and temporal filtering, but not by along-track filtering. It thus seems like a grave misinterpretation to say that the 1-D coherence reveals something meaningful about the along-track resolution.

I am going to focus most of this review on the first problem. Below, I give a general derivation that shows that, in a 1-D problem with noise and filtering, the coherence cannot provide a useful measure of resolution. I follow that with 1-D examples, with and without noise. Finally, I include some 2-D examples to illustrate the second problem listed above.

**1 The coherence method in 1-D**

To see that the coherence-based measure of resolution does not tell us anything directly about the filtering properties of DUACS mapping system, consider a 1-D mapping on a uniform grid. Let the measured SSH be $\tilde{x} = x + n$, where $x$ is the true SSH and $n$ is the measurement noise. We estimate the SSH as some linear combination of the measurements:

$$h_n = \sum_{m=-\infty}^{\infty} w_m \tilde{x}_{n-m}. \tag{1}$$

The weighting function $w$ specifies the particular linear combination, and could represent any linear mapping; the weighting function is assumed to be nonzero over only a finite extent, so the summation indices do not actually extend to infinity. Equation 1 is a convolution operation, which we write as,

$$h = w * \tilde{x} = w * (x + n), \tag{2}$$

where $*$ is the discrete convolution operator. Fourier transforming both sides and using the convolution theorem,

$$\hat{h}(k) = \hat{w}(k)\big(\hat{x}(k) + \hat{n}(k)\big), \tag{3}$$

where the hats indicate the Fourier coefficients. $\hat{w}$ is important because it specifies the filtering properties of the mapping– it is the filter transfer function that relates the Fourier transform of the mapped data to the Fourier transform of the input data. This follows directly from Equation 3.

In the paper, the mapped field is compared to independent data (data withheld from the mapping), which we will represent as $y = x + n_0$, where $y$ is the measured SSH and $n_0$ is the noise on that independent record. The Fourier transform of the independent SSH record is,

$$\hat{y}(k) = \hat{x}(k) + \hat{n_0}(k). \tag{4}$$

No filtering is applied to this independent record.

The squared coherence between the mapped SSH $h$ and the independent SSH record $y$ is,

$$\gamma^2 = \frac{\langle \hat{y}^* \hat{h} \rangle^2}{\langle \hat{y}^* \hat{y} \rangle \langle \hat{h}^* \hat{h} \rangle}, \tag{5}$$

where the superscript asterisk denotes the complex conjugate. To estimate the coherence and gain, we must estimate $\langle \hat{y}^* \hat{y} \rangle$, $\langle \hat{h}^* \hat{h} \rangle$, and $\langle \hat{y}^* \hat{h} \rangle$. They are given by,

$$\langle \hat{y}^* \hat{h} \rangle = \hat{w} \big( \langle \hat{x}^* \hat{x} \rangle + \langle \hat{n}_0^* \hat{x} \rangle + \langle \hat{x}^* \hat{n} \rangle + \langle \hat{n}_0^* \hat{n} \rangle \big), \tag{6}$$

$$\langle \hat{y}^* \hat{y} \rangle = \langle \hat{x}^* \hat{x} \rangle + \langle \hat{x}^* \hat{n}_0 \rangle + \langle \hat{n}_0^* \hat{x} \rangle + \langle \hat{n}_0^* \hat{n}_0 \rangle, \tag{7}$$

$$\langle \hat{h}^* \hat{h} \rangle = \hat{w}^2 \big( \langle \hat{x}^* \hat{x} \rangle + \langle \hat{x}^* \hat{n} \rangle + \langle \hat{n}^* \hat{x} \rangle + \langle \hat{n}^* \hat{n} \rangle \big). \tag{8}$$

(Note that we are assuming the mapping operation does not introduce a phase shift, and thus that $\hat{w}$ is real.) Now, if we assume the noise is uncorrelated with the true SSH and that the noise between different passes/instruments is uncorrelated, then,

$$\langle \hat{x}^* \hat{n} \rangle = \langle \hat{x}^* \hat{n}_0 \rangle = \langle \hat{n}_0^* \hat{n} \rangle = 0. \tag{9}$$

Then,

$$\langle \hat{y}^* \hat{h} \rangle = \hat{w} \langle \hat{x}^* \hat{x} \rangle, \tag{10}$$

$$\langle \hat{y}^* \hat{y} \rangle = \langle \hat{x}^* \hat{x} \rangle + \langle \hat{n}_0^* \hat{n}_0 \rangle, \tag{11}$$

$$\langle \hat{h}^* \hat{h} \rangle = \hat{w}^2 \left( \langle \hat{x}^* \hat{x} \rangle + \langle \hat{n}^* \hat{n} \rangle \right). \tag{12}$$

Now, we can substitute into the coherence expression to obtain,

$$\gamma^2 = \frac{\langle \hat{y}^* \hat{h} \rangle^2}{\langle \hat{y}^* \hat{y} \rangle \langle \hat{h}^* \hat{h} \rangle} = \frac{\langle \hat{x}^* \hat{x} \rangle^2}{\big( \langle \hat{x}^* \hat{x} \rangle + \langle \hat{n}_0^* \hat{n}_0 \rangle \big) \big( \langle \hat{x}^* \hat{x} \rangle + \langle \hat{n}^* \hat{n} \rangle \big)}. \tag{13}$$

If we assume that the noise in the input data has the same spectrum as the noise in the independent data ($\langle \hat{n}_0^* \hat{n}_0 \rangle = \langle \hat{n}^* \hat{n} \rangle$), this simplifies somewhat to,

$$\boxed{\gamma^2 = \frac{\langle \hat{x}^* \hat{x} \rangle^2}{\big( \langle \hat{x}^* \hat{x} \rangle + \langle \hat{n}^* \hat{n} \rangle \big)^2}.} \tag{14}$$

We can immediately see two important things: (1) when the noise variance equals the true SSH variance, the squared coherence has a value of 0.25 (not at 0.5 as assumed in the paper), and (2) the coherence is independent of the Fourier transform of the weighting function ($\hat{w}$), which is the

quantity that contains all of the information about how the mapping filters the data. This derivation clearly shows that, with or without noise, the coherence tells us nothing about the filtering in the 1-D example. Based on this, it should be impossible to infer anything about the smoothing in a 1-D example from the coherence. Note that, if either the noise on the independent data or the noise in the input data (prior to mapping) is assumed to be zero, we recover the situation assumed in the paper, in which the squared coherence is 0.5 when the SNR=1.

The transfer function between the mapped SSH $h$ and the independent SSH record $y$ is,

$$H = \frac{\langle \hat{y}^* \hat{h} \rangle}{\langle \hat{y}^* \hat{y} \rangle} = \frac{\hat{w} \langle \hat{x}^* \hat{x} \rangle}{\langle \hat{x}^* \hat{x} \rangle + \langle \hat{n}_0^* \hat{n}_0 \rangle}. \tag{15}$$

If the noise is small relative to the true SSH signal, the transfer function approaches the filter transfer function of the mapping. (If we used the actual input data instead of the independent, withheld data, the transfer function would be identical to the filter transfer function of the mapping.)

For completeness, we can also write down the ratio of the two spectra:

$$\frac{\langle \hat{h}^* \hat{h} \rangle}{\langle \hat{y}^* \hat{y} \rangle} = \frac{\hat{w}^2 \left( \langle \hat{x}^* \hat{x} \rangle + \langle \hat{n}^* \hat{n} \rangle \right)}{\langle \hat{x}^* \hat{x} \rangle + \langle \hat{n}_0^* \hat{n}_0 \rangle}. \tag{16}$$

If the noise in the input data has the same spectrum as the noise in the independent data ($\langle \hat{n}_0^* \hat{n}_0 \rangle = \langle \hat{n}^* \hat{n} \rangle$), this simplifies to a nice result,

$$\frac{\langle \hat{h}^* \hat{h} \rangle}{\langle \hat{y}^* \hat{y} \rangle} = \hat{w}^2. \tag{17}$$

The above derivation was developed for a 1-D situation, but it would come out exactly the same if we have a multidimensional mapping, a multidimensional Fourier transform, and compute the coherence against a multidimensional version of the withheld data. In the paper, there is only 1-dimensional data to compare to the multidimensional mapping, and this adds an additional difficulty related to problem (2) mentioned in the introduction.

**2  Examples to illustrate behavior of proposed measure of resolution**

I have prepared four examples to illustrate the behavior of the proposed measure of resolution and to compare it to more commonly used measures (the gain and the spectral ratio method). For the 1-D case, there are examples with and without noise. The examples also have no sampling errors. The examples thus focus on what the proposed measure of resolution can tell us about the filtering properties of the mapping algorithm. The four examples are: (1) a 1-D case without noise, (2) and 1-D case with noise, (3) a 2-D case with a white spectrum (no noise), and (4) a 2-D case with a red spectrum (no noise). In all four examples, I mapped the data using three different smoothers, a Gauss-Markov smoother (also known as optimal interpolation), a Gaussian weighted average smoother, and a quadratic loess smoother. At least a few of the authors are familiar with these smoothers, and the exact details of the smoothers are not important. The loess and Gaussian smoother parameters were chosen such that they filter the data with a 25-km half-power filter cutoff in one spatial direction (nominally the "along-track" direction). The Gauss-Markov

smoother uses a Gaussian autocovariance function and assumes the measurement contains a small amount of white noise, and it has similar but not identical filtering properties (with an autocovariance function spatial decay timescale of 25 km). In the 1-D example with noise, I also varied the filter cutoff wavelength to examine the sensitivity of the coherence to the filtering.

**2.1   1-D example, without noise**

In the first example, the input signal is a random realization of a process having a red spectrum ($k^{-2}$ power law), sampled on a uniform spatial grid. The input data were "mapped" (smoothed) to the same grid as the sampling positions. There is no measurement noise and no sampling error. In this case, the only thing limiting the resolution of the mapped fields is the filtering inherent in the mapping algorithm. One could imagine this as a case where there is a single along-track pass of data, and they have been mapped to a regular spatial grid (along-track).

Figure 1 (top) shows the spectrum of the input data and of the three mapped fields. The mapped fields have less variance at high wavenumbers because of the filtering inherent in the mapping.

The filtering inherent in the mapping is quantified more directly by computing the cross-spectral gain (or relative amplitude of variability coherent between the input data and the mapped data). The gain is equivalent to the magnitude of the filter transfer function of the filters, and is shown in Figure 1 (middle panel). The Gaussian smoother has a filter transfer function that resembles a Gaussian function (as is expected, because the Fourier transform of the spatially truncated Gaussian weighting function is approximately a Gaussian). The loess smoother has, as expected, a steeper filter roll-off (meaning it decreases from one toward zero more abruptly near the half-power point) with a small but noticeable filter sidelobe at about twice the half-power cutoff wavenumber (Schlax and Chelton, 1992, their Figure 1). There is another, even smaller sidelobe of the loess filter that has a maximum value near wavenumbers of 0.12-0.13. The Gauss-Markov smoothed estimate has a very steep roll-off, and it has a half-power filter cutoff that is at a slightly higher wavenumber than the loess and Gaussian smoothed fields. All of these same features can be clearly seen in the spectra, as well (upper panel of Figure 1).

Based on conventional understandings of the term "resolution", we would say that the resolution, as defined by the half-power point of the filtering, is highest in the Gauss-Markov mapping and is the same in the loess and Gaussian weighted average mappings. (One might reasonably argue the resolution of the Gaussian mapping is worse because it noticeably attenuates wavelengths longer than the half-power cut-off wavelength.) For example, this perspective is similar to the definition of resolution used by the SWOT project when they state the raw (downlinked) resolution of the in-swath SWOT data will be 1-km resolution and 500-m posting– the onboard processor will filter the data, which fundamentally limits the resolution.

Now, if we turn our attention to the squared coherence (bottom panel of Figure 1), and apply the authors' coherence-based definition of resolution, we would reach the opposite conclusion: the coherence with the Gaussian smoother has higher values at high wavenumbers, which the authors would interpret as superior resolution. The squared coherence is almost one, even at wavenumbers three times larger than the half-power point, where the variance is more than 100 times less than the actual (unfiltered) variance. Under the definition of resolution proposed by the authors, the resolution of the mapping would be 6.5km wavelength for the Gaussian mapping, 14.3km wavelength for the loess mapping, and 20km wavelength for the Gauss-Markov mapping. The maximum possible resolution is 6km wavelengths because the sample spacing in this example

[Figure]

Figure 1: 1D example without noise. Upper panel: Spectra of input signal and mapped fields. Middle panel: squared coherence of the mapped field with the input signal. Lower panel: spectral gain computed between the input signal and the mapped field. The black dashed line marks the theoretical half-power wavenumber of the loess and Gaussian smoothers.

is 3km.

The coherence in this noise-free example should have been one at all wavenumbers for all three mappings (Equation 14). I believe the reason the coherence is not one has to do with numerical errors (such as roundoff errors) that occur at the higher wavenumbers where the amplitude of the filtered signal approaches zero. Since the filter transfer function of the Gaussian has the slowest rolloff, numerical issues only gradually become a problem. They abruptly become a problem for the Gauss-Markov filter because its filter transfer function drops abruptly to zero through the filter cutoff wavenumber. The numerical issues for the loess smoother become problematic at the wavenumbers that correspond to nodes between the sidelobes, and less of a problem near the extrema of the sidelobes. The existence of two sidelobes can be seen in the figure. (Consistent with the idea that numerical errors are influencing the high-wavenumber coherence, I found that the coherence at the high wavenumbers changed when I did the following: (1) force Matlab to use a different FFT algorithm, which should affect the error of the FFT, and (2) reduced the number of points in the time series, which should decrease the error of the FFT. I suspect that most of the numerical errors are in the mapping.)

In summary, this first example shows that the coherence-based measure of resolution fails to be useful in a simple, but relevant, example. The example is relevant because the filtering of the mapping procedure is an important aspect of the resolution of the mapped field, and the coherence criterion tells us almost nothing about this filtering.

**2.2   1-D example, with noise**

This example is identical to the previous one, but independent realizations of random noise were added to both the independent data and to the input data. We did this for two different noise values– in one case the noise spectrum intersects the spectrum of the true SSH at lower wavenumbers than the filter half-power point, and in the other case the noise spectrum intersects the spectrum of the true SSH at higher wavenumbers than the filter half-power point. These cases are referred to as the high-noise and low-noise cases respectively. To make it easy to see how different filtering properties appear in the coherence, I also included two sets of filter cutoffs (25-km and 15-km wavelengths) for each noise case.

In the low noise case, the results are unsurprisingly very similar to the noise-free case (Figure 2). The spectral gain provides a qualitatively useful but quantitatively inaccurate description of the filtering, almost correctly identifying the half-power points and grouping the mappings into two clusters according to their filtering properties. In contrast, the coherence measure of resolution identifies resolutions of 20.6km, 14.8km, 12.3km, and has three mappings clustered around 9km (including both the 15 and 25km Gaussian mappings).

The high-noise case is qualitatively different than the low-noise case. In the high-noise case, all of the coherence curves tend to collapse onto one another (Figure 3), as expected from Equation 14. Also, the gain does not provide a very useful measure of the filtering. Because we have constructed this example so that the noise in the input data has the same spectrum as the noise in the independent data ($\langle \hat{n}_0^* \hat{n}_0 \rangle = \langle \hat{n}^* \hat{n} \rangle$), the spectral ratio method does well in both of these examples with noise, as expected from Equations 16-17.

[Figure]

Figure 2: 1D example, with "low noise". Upper panel: Spectra of input signal and mapped fields, including a partitioning of the signal and noise in the input signal. Middle panel: squared coherence of the mapped field with the input signal. Lower panel: spectral gain computed between the input signal and the mapped field. The black dashed lines marks the theoretical half-power wavenumber of the loess and Gaussian smoothers. Note that this figure uses a log scale for frequency, unlike some other figures.

[Figure]

Figure 3: 1D example, with "high noise". Upper panel: Spectra of input signal and mapped fields, including a partitioning of the signal and noise in the input signal. Middle panel: squared coherence of the mapped field with the input signal. Lower panel: spectral gain computed between the input signal and the mapped field. The black dashed lines marks the theoretical half-power wavenumber of the loess and Gaussian smoothers. Note that this figure uses a log scale for frequency, unlike some other figures.

**2.2.1  Summary of 1-D examples**

The coherence should be completely independent of the filtering and should only contain information about the SNR (Equation 14). In the 1-D example without noise, the coherence seems to indirectly contain information about the filtering, which appears to be related to the SNR associated with noise from numerical errors. In this case, application of the coherence method yields estimates of the resolution that range from 6.5-20 km for three different smoothers with nearly equivalent half-power wavelengths of about 25 km. In the more realistic cases that contain noise, the coherence method applied to the high-noise case (when the wavenumber at which SNR=1 is at a lower wavenumber than the filter cutoff) does give important information about the SNR in the input data and the independent validation data, but it tells us nothing at all about the differences in the filtering in the the six different smoothing scenarios. The coherence method yields no reliable information about the smoothing in the low-noise case (when the wavenumber at which SNR=1 is at a higher wavenumber than the filter cutoff). In this latter case, for example, the Gaussian smoothers with 25-km and 15-km half-power wavelengths are both identified as having about 9-km resolution using the coherence method.

The derivation and examples given above illustrate my basic objection to the coherence-based measure of resolution: it tells us nothing useful about the filtering in the DUACS mapping. It potentially can provide useful information about the SNR of the input data, but the SNR of the input data will be the same regardless of what mapping parameters are applied to the data. If the method is not robust in 1-D, we cannot safely apply it in 3-D.

**2.3  2-D example with a white spectrum**

We could really stop this discussion here. However, I imagine the authors might wonder, as I did when thinking about the above example, why the coherence-based measure of resolution seems to provide reasonable results when applied to the DUACS system. The next two examples show another problem with the coherence-based measure of resolution that is related to the fact that the filtering and spectrum in one dimension (e.g., time) has important effects on the coherence in the other dimension (e.g., space).

In this example, the input signal is a random realization of a process having a white spectrum, sampled on a uniform 2-D grid meant to represent space in one dimension and time in the other. The input data were "mapped" (smoothed) to the same space-time grid as they were sampled on. There is no measurement noise and no sampling error. In this case, the only thing limiting the resolution of the mapped fields is the filtering inherent in the mapping algorithm. One could imagine this case as a case where there are repeated along-track passes of data, and they have been mapped to a regular space-time grid (along-track). Alternatively, instead of thinking of one dimension as time, we could imagine the two dimensions as along- and across-track directions. (Please note that these examples have noisier and more coarsely resolved spectral estimates because I could not afford the computer power/time to use a very large number of samples.)

In this second example, the smoothing in the "along-track" direction is the same as in the 1-D example. In the other dimension (call it time), there is some smoothing (10-unit half-power wavelength for the loess and Gaussian smoothers, and similar for the Gauss-Markov mapping), but the exact value doesn't matter.

In analogy to the approach used in the paper, I took a single "along-track" sample of the input data and of the 2D mapped data (from the central "time" of the mapped domain, to avoid edge

[Figure]

Figure 4: 2D example for a variable that has a white spectrum. Upper panel: Spectra of along-track samples of input signal and mapped fields. Middle panel: squared coherence of the mapped field with the input signal. Lower panel: spectral gain computed between the input signal and the mapped field. The black dashed line marks the theoretical half-power wavenumber of the loess and Gaussian smoothers in the along-track direction.

effects in the mappings). The spectra of the along-track input data and mapped data are shown in Figure 4 (upper panel). The variance of all three mapped fields is reduced at all wavenumbers relative to the input data, but the variance reduction is greatest at the highest wavenumbers. This is easy to understand– it just reflects the fact that the along-track filtering affects the larger along-track wavenumbers but the temporal filtering affects all along-track wavenumbers equally and accordingly causes a uniform reduction in variance. (This is analogous to how the across-track smoothing of SWOT data should reduce the contributions of noise to the along-track spectrum.)

The estimates of gain between the along-track input data and mapped data tell a similar story (Figure 4, middle panel). The mapped fields are attenuated relative to the input data at all along-track wavenumbers because of the temporal filtering, and they are even more attenuated at the high along-track wavenumbers because of the along-track filtering.

The squared coherence is not especially interesting. Squared coherence in a particular wavenumber band can generally be interpreted as the fraction of the variance at that wavenumber in the input record (the raw along-track data) that can be accounted by multiplying the Fourier transform of the output record (the along-track mapped data) by some complex-valued constant (the value of the transfer function at that wavenumber). So, we can see another aspect of what we saw in the spectra and gain plots– the coherence is reduced at all wavenumbers because the temporal filtering reduced the variance in the mapped fields at all wavenumbers relative to the raw along-track record, and thus there is only limited ability of the along-track mapped data to account for the variance in the raw along-track data. (I struggled to understand this, but I found this simple example helpful: imagine if the temporal averaging were very extreme such that the mapped data is very close to the time-mean SSH anomaly; in that case, we would expect very low along-track coherence with an along-track pass of SSH anomaly that would be dominated by eddies and variability.)

If we did try to apply the coherence-based measure of resolution in this example, we would conclude that the wavelength resolution of the Gauss-Markov and loess mappings is larger than the domain size (1800 km), and that the Gaussian mapping might resolve about 50 km wavelengths.

**2.4   2-D example with a red spectrum**

I imagine the authors still might wonder, as I did when thinking about the above example, why the coherence-based measure of resolution seems to provide reasonable results when applied to the DUACS system. The coherence in Figure 1 of the paper, for example, does not look like the coherence in Figure 1 or in Figure 4. Instead, Figure 1 of the paper has high coherence at low wavenumbers and low coherence at high wavenumbers. I think the explanation is that the variabilty in SSH has a red spectrum in space and time, so that the low wavenumbers tend to be associated with energetic low frequencies, and the low frequencies tend to be associated with energetic low wavenumbers. This example, with a signal that has a red spectrum in both wavenumber and frequency, is meant to illustrate how the coherence-based measure of resolution applied to a red signal spectrum combined with the multiple-dimension filtering of the DUACS system can lead to results that seem reasonable, even though the apparently reasonable looking results are basically accidental.

In this example, the input signal is a random realization of a process having a red spectrum in both wavenumber and frequency (spectrum proportional to $k^{-2}\omega^{-2}$), sampled on a uniform 2-D grid meant to represent space in one dimension and time in the other. All other aspects are

identical to the second example.

I again took a single "along-track" sample of the input data and of the 2D mapped data. The spectra of the along-track input data and mapped data are shown in Figure 5 (upper panel). Unlike the second example with white spectra, the variance of the three mapped fields is not noticeably reduced at low wavenumbers. This is not difficult to understand– there is relatively little variance at the high frequencies in the temporal dimension, so the temporal low-pass filtering associated with the mapping has relatively little effect on the variance in the along-track direction.

The gain plots in this example (Figure 5, middle panel) look more similar to the 1-D example than to the 2-D example with a white noise signal. The half-power points one would infer from the gain are at similar wavenumbers to the theoretical 25km half-power point of the loess and Gaussian smoothers (29km wavelength for the loess, 27km wavelength for the Gaussian, and 22.7km wavelength for the Gauss-Markov mapping). This good agreement is totally accidental. If the frequency spectrum of the SSH were different, or if the temporal smoothing were different, the gain between the along-track input data and mapped data would change, and the point where it is equal to $\sqrt{0.5}$ (i.e., the half-power point) would be different.

The squared coherence in this example looks qualitatively similar to Figure 1 of the paper, with high coherence at low along-track wavenumbers and low coherence at high wavenumbers. The coherence-based definition of resolution would yield along-track wavenumbers of 8.5km wavelength for the Gaussian mapping, 14.7km wavelength for the loess mapping, and 19.9km wavelength for the Gauss-Markov mapping.

**3 Conclusion**

The along-track filtering properties of the mapping schemes should be the same in all three examples. (For example, we can analytically derive the filtering for the Gaussian weighted average.) The half-power point of the filtering would not be a good specification of the resolution, in general, because the resolution will also depend on the sampling and on the SNR of the input data. However, in these examples, which do not have any noise or sampling errors, the effective resolution should be determined only by the filtering. The filter half-power points of the along-track filtering were at a wavelength of about 25km for all three mapping schemes.

I suppose all measures of resolution have their drawbacks (e.g., the spectral ratio approach discussed in the paper is subject to some of the same issues, such as sensitivity of inferred along-track resolution to temporal filtering), but I do not see any theoretical basis for the coherence-based measure of resolution. In the cases we considered, the coherence-based measure yielded along-track wavelength resolutions ranging from 6.5km (close to the Nyquist wavelength) to >1800km (the domain size) for cases in which the actual resolution was about 25km, and I am confident that almost any result could be obtained by varying the signal spectrum and/or the temporal filtering. In a case with no noise and no sampling errors, the along-track wavenumber resolution should not depend on the signal spectrum or the temporal filtering. I cannot think of any relevant case in which it would make sense to define the resolution as the wavenumber where the squared coherence between the mapped product and a one-dimensional sample of the input data is equal to 0.5.

I think that the conclusions of the paper are not quantitatively useful and that it would be counterproductive to publish this. I feel really bad having to say that, because it is clear the authors worked hard to make a high-quality paper. Except for the one methodological flaw, it is

[Figure]

Figure 5: 2D example for a variable that has a spectrum that is red in wavenumber and frequency. Upper panel: Spectra of along-track samples of input signal and mapped fields. Middle panel: squared coherence of the mapped field with the input signal. Lower panel: spectral gain computed between the input signal and the mapped field. The black dashed line marks the theoretical half-power wavenumber of the loess and Gaussian smoothers in the along-track direction.

an excellent paper in all other respects.

I have focused on the fact that the coherence-based measure of resolution does not take into account the filtering by the mapping system. The coherence-based measure of resolution also does not adequately take account of the sampling, and this might also be a serious issue.

**Acknowledgements**

I thank Dudley Chelton for constructive critical comments on two earlier versions of this review.

---

## Author Response (AR2)

**Reponse review #2 Tom Farrar**

The authors have improved the manuscript dramatically in their revised version. I believe the new analysis method is sound, and the description of what was done is also much clearer. It seems like a substantially different paper than the previous version I reviewed, and I am happy to recommend publication after minor revisions. I would also like to thank the authors for their pleasant and constructive attitude in responding to my previous, critical review. (There were actually two versions of the previous review-- I made an updated one with more detail and gave it to the authors directly, but the OSD system would not allow me to upload that review because the online discussion period had ended. I am uploading that review now.)

My only substantial reservation about this version of the manuscript has to do with the approach and conclusion about the temporal resolution. Something does not add up, as described in the following paragraph. I suggest the authors investigate this a little more and either add this as a caveat and change the plot and conclusions accordingly or adjust the analysis to have finer frequency resolution.

**Authors response (AR): We would like to thank Tom Farrar for the review and the suggestions he made on this new version of the manuscript. We provide below the answers to the two main concerns he raised.**

Page 4, lines 26+: There is another important consequence of choosing to analyze 100-day segments. This sets the coarseness of the resulting evaluation of temporal resolution, by setting the frequency resolution of the spectral analysis. For 100-day segments, the frequency resolution is 1/100 cycles per day (cpd) (actually, the frequency resolution is effectively somewhat less because of the taper window, but I think it is the frequency grid that matters here). If the record chunks are actually 100 days long as stated (instead of 100 data points long), the frequency bands will be centered at frequencies corresponding to periods of 101, 50.5, 33.67, 25.25, 20.20, 16.83, and 14.43 days. This means that the analysis should only be able to coarsely determine the temporal resolution (e.g., 50.5 days versus 33.67 days). It seems to me that the color scale chosen for displaying the results is thus misleading, or else, there has been some kind of interpolation of the spectral results to arrive at values between 34 and 50 days. It is not clear to me that such interpolation would be a valid approach.

AR: We agree that the choice of 100-day long segment set the coarseness of the resulting evaluation. In our analysis, we set the segment length to 100 days long to get many realizations. It also corresponds to 100 continuous data points for both tide gauge and map datasets. In other words, segments that are shorter than 100 data points (100-days long) are excluded from the analysis. Consequently, analysis at some tide gauges are lost when choosing longer segment sizes. In total, we found that 2 sensors are lost when choosing 300 days long-segments, 3 sensors when choosing 1 year-long segment. When choosing 2-year long period, we lose 20 sensors. Since the number of lost sensors between 100-days to 365

days periods not significative (for a total of 190 tide gauges datasets), we propose to use the analysis performed over segment that are 365-days long to get finer spectral resolution. We modify the manuscript accordingly (key value and figures 3 and 5).

Additionally, we attach below the results to a sensitivity study we performed for 100 data points (100 days) long segment, 200 data points segments (200 days) and 365 data points segments (1 year). As for the estimation of the spatial scales, we linearly interpolate between successive NSR values to find the wavelength corresponding to the NSR of 0.5. This linear interpolation indeed explains the value found between frequency scales. We illustrate that the segment length has a weak impact on the estimation of the effective temporal resolution from 40° to the poles. Larger difference is found in the inter-tropical band (cf. Fig.4 below: zonally averaged time scale for various segment size). Figure 1 to 3 show the maps of estimated effective temporal resolution using 100 days long segment, 200 days long segment and 365 days long segment. The spatial distribution of the temporal resolution looks relatively similar between each test. To illustrate the differences, we plot the zonally averaged effective resolutions for each segment sizes in Figure 4. The largest difference take place in the inter-tropical band.

Test#1: segment length = 100 days corresponding to frequency (in days):

Fig. 1 : Effective temporal resolution computed from 100 days long segment

Test#2: segment length = 200 days corresponding to frequency (in days):

200.0, 100.0, 66.7, 50.0, 40.0, 33.3, 28.6, 25.0, 22.2, 20.0, 18.2, 16.7, 15.4, 14.3, 13.3, 12.5, 11.8, 11.1, 10.5, 10.0, 9.5, 9.1, 8.7, 8.3, 8.0 ...

Fig 2.: Effective temporal resolution computed from 200 days long segment

Test#3: segment length = 365 days corresponding to frequency (in days): 365.0 182.5 121.7 91.2 73.0 60.8 52.1 45.6 40.6 36.5 33.2 30.4 28.1 26.1 24.3 22.8 21.5 20.3 19.2 18.2 17.4 16.6 15.9 15.2 14.6 14.0 13.5 13.0 12.6 12.2 11.8 11.4 11.1 10.7 10.4...